# Exploring the Dynamics between Specialization and De-Specialization of Welfare Experiences: A Qualitative Study of the Special Families under the One-Child Policy in China

**DOI:** 10.3390/ijerph20054327

**Published:** 2023-02-28

**Authors:** Sheng-Li Cheng, Miao Yu, Shu-Shan Liu, Yun Li

**Affiliations:** Department of Social Work, School of Philosophy and Social Development, Shandong University, Jinan 250100, China

**Keywords:** special families under one-child policy, welfare experiences, qualitative research, specialization, de-specialization

## Abstract

The one-child policy, i.e., of having only one child per couple, was adopted as the essential family policy in China from 1979, and since the beginning of the 21st century, it has given rise to problems of special families under the one-child policy caused by the death or disability of only children. The existing research focused on the issue of special families from a macro-social level and analyzed the welfare demands and welfare policies of those families, whereas less research has been concerned with the families’ individual experiences and interpretations. This study adopted a qualitative research method and conducted in-depth interviews with 33 participants to analyze the welfare experiences of special families in Jinan city, Shandong Province. The findings of the study were based on generalized analyses of the interviews, including the “specialization” dimension of welfare experiences with identity-oriented, targeted, and comprehensive characteristics, the “de-specialization” dimension of welfare experiences with identity-denied, excluded, and hidden characteristics. The dynamics between the two dimensions among different special families, different family members, and different periods in the families’ lives were also examined. We present a discussion of the study’s findings and their implications, categorized into the theoretical and practical domains.

## 1. Introduction 

The one-child policy, i.e., of having only one child per couple, has been the essential family policy in China since its adoption in 1979, and it has effectively controlled the trend of rapid population growth in China. The policy has also significantly influenced China’s economic and social transformation through the nation’s progress of urbanization and modernization. However, the opposite side of this policy has also been apparent since the beginning of the 21st century [1] and has given rise to the problems of *special families under the one-child policy* caused by the death or disability of families’ only children. In current policy and regulations, the National Population and Family Planning Commission of China defines *special families under the one-child policy*, or, in another phrase, “the family planning policy”, as those families whose only child is disabled or deceased and who have not given birth to or adopted another child. This study used the concept of “special family” according to the official definition raised by China’s Health and Welfare Commission, rather than another popular definition––the “Shidu family”, an expression used in academic areas to refer to a family that has lost its only child––because the concept of special family includes and is broader than the “lost-single-child family”, and not only the “Shidu family” but also the families in which the only child has a severe disability are targeted as recipients of some special welfare policies, and the concept of the special family serves better if it involves those who have unique experiences and interpretations about the implementation of these welfare policies, while we also used the concept of “Shidu family” sometimes in the article according to the context. Some scholars estimate that the number of special families will reach 4.5 million by 2050 [2], while the issue of how to provide them with the necessary support services and solve their difficulties caused by the loss or disability of their only child has gradually become an unavoidable welfare concern for the authorities and the society as a whole. 

In 2021, the Chinese government introduced a policy allowing couples to have three children, this may solve the problem of low fertility rate to a certain extent, and eliminate the possibility of special families in the future; however, we believe that the three-child policy will not diminish the importance of special family research, but make it more realistic and urgent, because the special families are now at risk of being ignored and forgotten by the society. In 2006, the State Council of the Central Committee of the Communist Party of China (CPC) issued a decision on “Comprehensively Strengthening Population and Family Planning Work to Solve Population Problems”, which explicitly proposed to actively explore the establishment of a support system for families with disabled and deceased only children. On the basis of that decision, in 2007, the National Population and Family Planning Commission and the Ministry of Finance issued the “Pilot Program of National Support System for Families Whose Only Child Died or is Disabled”, officially implementing a pilot program that established a support system for special families in some provinces and soon spread throughout the country. Since then, the social support system for special families has been continuously improved, with the contents of the social support being developed from an initial, limited financial compensation and material support to the inclusion of psychological support and spiritual consolation, and with its provision of social support gradually being institutionalized and systematized [3,4]. However, a large concern remains that the welfare demands of special families are not being fulfilled effectively––specifically, that the standards and level of support are a little low, the supply of social support services is insufficient to cover all of the special families in need, and the capacity of social support services is not able to satisfy the demand that special families have for professional social support services [5]. In response, the primary policy recommendations have called for increasing the investment of support funds, raising the standards and level of support, and increasing the number of personnel and the professional capacity of the social support services [6]. 

The existing research focused largely on the issue of special families from a macro-social level, with fewer studies having been concerned with the experiences and interpretations of those families. Our research is a qualitative study that we conducted in Jinan city, Shandong Province, in which we attempted to explore and analyze the subjective welfare experiences of the parents in special families. The research question is: what is their welfare experience and how do they interpret it, being the recipient of the special family support policies which were developed specially for them? According to our findings, the special families’ support system meets their needs for social support in many ways, but there are still a large number of cases that do not receive the social support they deserve. Some special families refuse to accept certain support provided by the government and society because they feel the services are presented negatively toward their special status, while other families insist on special support arrangements because of their special status, and some of them even insist on special support activities that go beyond what the government and society can provide. From our qualitative analysis of our in-depth interviews, the participants’ experiences and interpretations of achieving related welfare policies are shown to be complicated, and we have attempted to explore and understand the complications. On one hand, these special families need to be provided with “special” services, but on the other hand, they are still ordinary older people who also need “general” and “de-specialized” services. Thus, the dynamic between the specialization and de-specialization dimensions of the services approach poses a serious challenge to the authorities’ efforts to provide welfare services for special families.

## 2. Literature Review

Although other countries also have special hardship families formed by older parents who have suffered the death or disability of their children, and there are some researches on families who lost a child which mostly focused on issues such as poor physical and mental health [7], low economic well-being level [8], and high subsequent fertility [9], special families under China’s one-child policy are a unique social problem because the Chinese government has strictly implemented the family planning policy of only one child per couple for more than 30 years. The general two-child policy implemented in 2015 signaled the end of the “one-child policy” in China, which made many one-child families, especially the special families, feel that they had become victims fooled by the one-child policy and were about to be forgotten by the welfare system. This led to some collective rights protection actions of special families with the aim of obtaining welfare support from the government, which aroused the high attention and concern of government and society [10]. In addition, in just the past 20 years, the number of special families has considerably increased, which has attracted attention from academicians, who have mainly focused on the welfare demands of special families and related policies as coping strategies.

### 2.1. The Welfare Demands of Special Families under the One-Child Policy

From the existing knowledge, the analysis of the welfare demands of special families has been presented as a process of development from one simple dimension to a situation with multiple dimensions, and from focusing on material needs to attending to spiritual needs. Researchers gradually have clarified that the needs of special families are multifaceted, multidimensional, and differentiated, rather than being just a simple need for homogeneous economic or material support. Specifically, studies have found that the older parents who had lost their only children asked not only for financial compensation, but also for spiritual comfort [11,12]. The death of an only child brings unbearable psychological trauma to the family, creating needs both for medical services and for spiritual consolation [12]. The social support needs of the special families comprise not only financial compensation and material support, but also social support and old-age care, and the social support needs of the special older adults are different [13]. The vulnerability of the special older adults is manifested in various respects, and their demands are all-round [14]. The social support needs of the special older adults are affected by their gender, age, household registration, marital status, class structure, health status, reasons for loss of independence, and the presence or absence of their third generation [15,16].

Scholars paid attention to the differences in the social support needs of special families in urban versus rural areas and determined that their demands for older individuals’ care services tended to increase from rural areas to general urban areas and then, to more developed cities [17]. Rural special families have the highest level of demand for health care services, followed by material and economic needs, followed by spiritual and recreational needs, and then by needs for life care services [18]. Current research analyzed the welfare demands in special families, in terms of targeted needs, such as the need for psychological guidance due to grief over the loss of their child [11,12,19] and the need to strengthen their social capital in response to the breakdown of their social networks [20]. The existing studies about the demands of the special older individuals emphasized their targeted needs as older people without children, and paid less attention to their general welfare needs. Thus, the research findings based on the literature and second-hand studies, with rare empirical studies, raise the urgency for further exploration of the experiences of these special families. 

### 2.2. The Welfare Policies for Special Families under the One-Child Policy

In conjunction with the evolution of the welfare demands of special families under the one-child policy, the related welfare policies for special families have also undergone a transformation, from focusing only on financial compensation and material support to providing the families with diversified, stratified, and differentiated types of social support. Some studies focused on how to provide the necessary economic support and social security for special families under the one-child policy [21], while other scholars gradually stressed the importance of establishing a more comprehensive social support system for these special families and have emphasized the need for an economic relief mechanism for them [12,22]. Scholars highlighted that these special families are an unintended consequence of one-child related policies and should receive comprehensive welfare support in response to their needs [19]. In order to provide for the special families’ diverse welfare demands, some scholars suggested the establishment of a “five-in-one” mechanism for social support, involving a collaboration of the government, the family, the market, society, and the community [17]. Another group of scholars emphasized that the government should take the lead in establishing a “caring community” model that can build a more compassionate and caring social environment by integrating and directing various public and social resources [23]. Other scholars expressed the concern that interventions should instead focus on age-related issues (e.g., medication, hospitalization, daily care) [24]. It was demonstrated that existing support policies improved the living conditions of special families to some extent [25], while it was also suggested that special support policies overemphasize the special nature of loss of independence, which may have a labeling effect and even lead to extreme cases of over-reliance or non-use of the policies by special families [26].

### 2.3. The Narratives of Special Families under the One-Child Policy

In addition to investigating the welfare demands of special families and the related policies that exist for them, we reviewed studies on the subjective experiences of special families that focused primarily on those families’ identity and marginalization. The sudden withdrawal of their family role at the loss of their only child made the parents unable to orient themselves and created temporary barriers to emergency interactions [27]. Furthermore, the changes in the parents’ family structure caused their long-established family identity to disappear and their self-cognition to appear to be chaotic [28]. The Shidu status is a new starting point for special families in their effort to rebuild themselves, and it can become their main identity [29]. Studies highlighted that some special families are more resistant to their special Shidu identities, however, reporting that those respondents give avoidance-type responses to collective emotional compensation for special family groups and to status-targeted benefits such as early retirement [30,31,32]. 

In the studies of marginalization of special families, some scholars suggested that their marginalization includes both structural and psychological dimensions [29], and that their individual marginalization follows a psychological-structural path in which the special families gradually form a psychologically marginalized identity that changes their thoughts and actions in social interactions, and ultimately affects their interactions with the outside world and individuals, leading to structural marginalization [31]. From the perspective of psychological marginalization, they are desperate and self-isolated due to the influence of traditional cultural concepts such as ‘many children, many fortunes’ and ‘four generations in one family’, and they actively separate themselves from their normal families [33,34]. They may also reject sympathy and compassion from others, manifesting themselves as being “disassociated” from friends and community interactions [27]. From the perspective of structural marginalization, special families gradually dissolve their original social interaction system due to the denial of self-worth [27,31]; they gradually develop inward-looking interactions, form in-groups, and consciously alienate themselves from the general group [27,32]. The dramatic decrease in social interaction due to low emotional energy and the accumulation of a large amount of negative emotions contribute to their pathological reputation, which easily invites social rejection [31]. Scholars also noted that the mindset of avoiding bad luck led to the exclusion of special families by normal families, reinforcing the structural marginalization of this group [35]. Some scholars are concerned that the existing studies, with their “suffering narratives” and “social redemption narratives” as the main narrative approach, have labeled special families as stigmatized, thus strengthening their marginalization at the social structure level [36].

## 3. Research Methods

This study adopted an interpretivist paradigm and qualitative research method to explore the welfare experiences of the parents who lost their only child, and we attempted to present the subjective narratives and the inductive qualitative themes of welfare experiences of those special families. This study belongs to a program of “promoting the service system for special families under the One-child Policy” delegated by the National Health Commission. The research team conducted their empirical qualitative study in Huaiyin district, Jinan city, performing 33 in-depth interviews in January and February 2022. Jinan city was chosen as the study site mainly because it is a pilot site of special families welfare policy, which was awarded “the national Warm Heart House Demonstration Site” by the authority; moreover, it is also the location of the study team which provided convenience for conducting the study.

### 3.1. Data Collection

In this study, 33 interviewees were selected by following the purposive sampling method, as we sought to gain a comprehensive and deep understanding of the welfare experiences of special families under the one-child policy. Sixteen participants from special families were selected by following the sampling criteria including: being a citizen living in the Huaiyin district, they have lost their only child. The semi-structured in-depth interviews were conducted, and the questions were centered around the following subjects: the special families’ welfare demands for community support services; their status of satisfaction with whether their demands were met, and their further requirements for support services; the current situation, experiences, difficulties, and challenges faced by communities in carrying out support services for these special families. Additionally, the other 17 were providers of related welfare services who could elaborate the welfare experiences of the special families and enrich the understandings of the researchers. They were selected by following the purposive sampling of recruiting the direct welfare providers under the current welfare system, including managers of a planning family support committee for the streets and community, full-time or part-time social workers in the community, related staff members of the District Civil Affairs Bureau, and the person in charge of special family support from the National Health Commission. The semi-structured in-depth interviews were also conducted, the questions focused on the opinions on the current welfare system and policy suggestions for improving support services in the future. The codes of the interviewees and related specific information are shown in Table 1. The study took the appropriate ethical considerations, including requiring all participants to complete an informed consent form, following the “no-harm principle”, and keeping all of the participants anonymous. All of the interviews were audio-recorded and were fully transcribed; each interview lasted 1–2 h.

### 3.2. Data Analysis

The qualitative data analysis software NVivo 12 was used for our qualitative interview analysis, and among other things, it provided inductive themes that emerged from the data analysis. The NVivo 12 software has powerful coding, summarization, and visualization functions that are especially suitable when the number of textual materials is large, and that helped to improve the accuracy and efficiency of the analysis and ensured the objectivity and truthfulness of the research findings [37,38]. On the basis of the purpose of the research, which attempted to analyze the welfare experiences of the special families, the grounded theory qualitative method was adopted as the data analysis method. We conducted the grounded theory analysis following the guidelines laid by Corbin and Strauss [39] which included three stages: open coding, axial coding, and selective coding. Specifically, the coding process of this study was divided into the following three steps. (1) First, at the open coding stage, materials from the 33 interviews were imported into the software, the original materials were carefully read and open coded, and the interview materials were open coded as comprehensively as possible, forming 59 free nodes and 1043 information reference points related to the welfare experiences that the special families reported with regard to family planning. (2) Next, at the axial coding stage, the open codes were summarized and the logical relationships were extracted to form summarized categories, such as specific welfare demands, reliance on the government, excluding themselves from normal life, feeling afraid of “specialized treatment”, and so on. (3) Then, at the last selective coding stage, we conducted a selective and theoretical analysis for the existing categories, and we attempted to present a deep interpretation and reflection of the interviews and build inductive theoretical themes: the deployment of the specialization and de-specialization dimensions of welfare experiences. The dynamics between those two dimensions were also discussed.

## 4. Findings

### 4.1. Specialization of Welfare Experiences: A Dimension Characterized as Being Identity-Oriented, Targeted, and Comprehensive

There is an old saying in China that refers to “yang er fang lao (raising children to support one in one’s old age)”. The death of the only child, who was supposed to grow up and assume many family functions, such as supporting the parents when they are old, carrying on the family line, and providing financial support for the family, has an irreversibly negative impact on the structure and function of the family. Compared with ordinary families, the special families in our study tended to need special care in order to maintain a normal life, and we defined those needs as the “specialization” of welfare experiences, which were shown to have the characteristics of being identity-oriented, targeted, and comprehensive.

China’s one-child policy is said to be “the boldest and largest experiment in population control in the history of the world” [40], meaning that the special status of special families was created in a specific social context. In the context of social identity, an individual’s psychological status is closely related to his or her family members’ status [41], and special families tend to define their self-perceptions and welfare needs on the basis of their special Shidu identity status.

The worry of our Shidu families was that “if something else happens, there is nothing we can do, we have [no] child to rely on. To put it in a bad way, we lost our family, [next] is to rely on the Communist Party, I hope [it will] care more about our(?) care. Auntie Lu (alone) called two days ago and said that I still have a big brother to take care of me, but for her, there is really nothing she can do if something happens. This is the biggest worry”. (Interviewee DDJF-4).

In the temporal dimension, the welfare experiences of special families exist in different forms that have different impacts on their identity construction at different life stages, with traces of transformation and continuity constructed by society during their life course [42]. Taking the loss of their only children as the starting point, the special families continued to play socially prescribed roles and practices in their subsequent life course, which can be divided into the early stage, which is characterized by self-isolation and sensitivity (in which the special family is dominated by emotions), the middle stage, which is characterized by despondency and requests for changes (the first exploration of social support for the special family), and the late stage, which is characterized by a reestablishment of order (the structure and function of the special family is restored, to some extent).

The early stage of Shidu: We also have this, [which] is a kind of severe depression, people simply do not want to see people, so we can only [go] through his relatives and(?) friends to understand his recent situation. (Interviewee DDJD)

The middle stage of Shidu: It’s definitely not the best, but it’s definitely necessary at first. You first have to have a process that makes them want to get out of the house, and after they get out of the house, you can only integrate [them] into normal activities if they see themselves as normal people. (Interviewee DQZF-1)

The late stage of Shidu: After the first phase of the project, the older individuals saw you with tears in their eyes, but now they are laughing and happy to see you coming from far away. In the past, they used to watch the performance on stage, but now they are performing on stage for you. Before, he was waiting for others to help him, but now he comes out to help others. (Interviewee DQZF-2)

The trauma of losing an only child or having a disabled only child cannot be healed, and the welfare experiences of special families permeate all aspects of their lives and remain with them forever. In sorting through the data, we found that the welfare demands of special families related mainly to economic support, psychological counseling, medical care, ageing care, and daily life care. The welfare needs of special families were twofold: the comprehensive needs of ordinary older families, and also their own unique targeted welfare needs. According to the data, special families varied in age, health status, and economic level, and they were influenced by multiple factors and had different combinations of needs. Differentiated needs are the source of precise welfare experiences, whereas an adequate connection between the supply and demand sides is a necessary guarantee for subjective welfare experiences [4], thus requiring comprehensive and also targeted services from the government and social organizations.

Old age… I think you are able to do with what you have, we can move, he can send us food at noon, it is okay. The other [issue] is the future needs, the main thing is that we don’t know how to pay for doctor’s appointments, and we don’t know how to escort people. Although I have this child, he is still small. The last time I almost called 120 at night, I didn’t dare to call 120, I had money, but no one taught me how to pay the bill. (Interviewee NXJF-3)

The second year after the death of the child, this thing hit really big, at that time [I] was as numb as if this thing [was] not true, but after six months and a year, our pain came, just think [ing] of the child [and] those things, feel [ing] so regretful, if anyone can forget? (Interviewee KSJF-3)

### 4.2. De-Specialization of Welfare Experiences: A Dimension Characterized as Being Identity-Denied, Excluded, and Hidden

As we mentioned earlier, the loss of a family’s original identity as parents brought deep and intense grief that lasted for a long time, and that persistent grief had a serious and negative impact on the physical and mental health of family members. From our analysis of the interviews, we found that the loss of the only child forced special families to face the deconstruction of their original identity and the reconstruction of their existing Shidu special identity. As we discussed above, the special families asked for special and targeted welfare when they were in their special-identity-oriented stage, and it was not expected that they could reconstruct and accept that identity peacefully. Identity-denied, excluded, and hidden characteristics also emerged from the subjective welfare experiences, and we defined this dimension as “de-specialization”. 

There are significant differences among the acceptable ways and outcomes of identity reconstruction in grief [19], and different special families have contrasting attitudes toward and perceptions of reconstructed special identities. The special families are regarded as similar to the “five-guarantee households” and the disabled groups that are included within the government’s list of the most vulnerable groups in need of policy support at present in China. The label of Shidu has rashly classified families from different backgrounds into one special category, and the identity of a disadvantaged group is not something that all special families can openly accept.

After my husband died, I was left on my own at home. My brother would sometimes come over to take care of me. I can’t walk very well, and I don’t participate [in] many activities, so I find it quite troublesome. I have not been in contact with people [for special families] before, but I may have had two phone calls with Xiao Chen (social worker). (Interviewee KSJD-4)

The traditional Chinese Confucian culture suggests that “there are three forms of unfilial conduct, of which the worst is to have no descendants”, and in ancient times, the premature death of a child was also associated with some taboos and curses in feudal thought [43]. That stigmatized ideology has been preserved in the course of social development, and in community life, that cultural tradition influences the interpersonal interactions of special families with their friends and neighbors. A social worker who provided services for special families mentioned: “He was in city L before, but now he has moved to city H. The reason he moved here is that he wanted to leave his familiar community, and after he came here, no one knew he is a person who lost his only child”. 

During the interviews, some special families were satisfied with their current life and the benefits they had received and did not ask for additional special attention. Differences in the levels of needs were present within the overall group of special families. On one hand, a low level of needs existed objectively for some and was influenced by the economic status, life background, and family structure of some of the special families. On the other hand, a high level of needs of special families can be hidden, with the causes of a “hidden high level of needs” being explained from the individual’s and society’s perspectives. From the individual’s perspective, special families tend to have low self-esteem, and the welfare system provides support for their retirement and livelihood, while on the spiritual level, it also carries the burden of worrying about bothering others. From society’s perspective, the culture of the community and the services used to help special families are closely related to the welfare experiences of special families. The emphasis on special status that staff and providers sometimes promulgate raises the risk of creating a secondary victimization for special families, whose dissatisfaction with the welfare that is provided can gradually cause them to tend to hide their true needs and marginalize themselves.

Some years ago, some social organizations signed contracts with community committees to organize regular activities for us to participate in, such as spring excursions, crafts, etc., and also organized us to visit their agencies. Those activities were so targeted that only our special families participated, and gradually no one want [ed] to go. (Interviewee DDJF-1) 

### 4.3. The Dynamics between Specialization and De-Specialization in Different Special Families, Family Members, and Family Life-Course Periods

In the process of providing support and assistance to special families, an overemphasis on the unique dimension of being special families as the whole unit, and a failure to account for the differences among different special families and different family members, can play the role of “labeling” and can lead to the two extremes, in which the families are either overly dependent on welfare and or have become welfare resistant [26]. By examining the interview data, a dynamic between the specialization and de-specialization dimensions of welfare experiences, as presented above, occurred within different special families, and also across the life courses of the family members.

#### 4.3.1. The Complexity of Welfare Demands among Different Special Families and Family Members

All special families experienced the trauma of their only child’s death and suffered the ensuing irreversible shocks and traumas in their family structure and functioning. However, as we have noted, significant differences also existed among their welfare experiences, according to our study, and the welfare demands of special families can be divided into two groups: one group of families that are welfare dependent and another group that are welfare resistant. The welfare-dependent special families regarded being special families as the outcome of a collectively created traumatic event during the implementation of the one-child policy, and they believed that the support they lacked due to the loss of their only child should have been compensated by the government and related services. In recent times, due to the general aging of the population, China is no longer limiting the nation’s population growth through the fertility policy, and the one-child policy is gradually receding from the historical stage. In the interim, special families continue to make their voices heard through petitions and other means, hoping to receive attention and obtain a version of targeted welfare.

We also found that, nationwide, these people were actually a hidden or potential group of petitioners, who felt they responded to the national one-child policy by themselves: “Now, I am not given any other protection policy because of the loss of independence. What about my next generation, the third generation? What will happen to them if they are gone?” Therefore, they have created a kind of petitioning group. (Interviewee DQZF-1)

In contrast, welfare-resistant special families believed that their needs for welfare services were similar to those of the general older adults, with needs in common that included medical care, ageing care, caregiving, and psychological comfort. Thus, different special families have different welfare needs, just as other ordinary older people do. Special families are ordinary families who have had the misfortune of losing their only children, and they may need some special support to help them through a tough time, but ultimately, they need to go on with their normal daily life. In the process of implementing the special family welfare policies, an overemphasis on their specialness as a group not only ignores the differences among them, but it also makes it harder for them to live their life as ordinary older people, and that can be the cause of their resistance. 

Community workers (one of whom was coded DDJC in the study) considered the concerns of welfare-resistant special families in their services and in planning activities for them, and the workers optimized their support methods without treating special families differently.

I never said this is an event for these families specially, I always said that our residents are invited to come. I never tell them that it’s for “Shidu” families, but [that] it’s a warm heart house, or from our family planning association, or a little something from our street, so they won’t be uncomfortable coming. (Interviewee DDJC)

#### 4.3.2. Life Course Approach to the Special Families

We found that special families often gradually shifted from the specialization-characterized welfare attitudes to de-specialization characterized welfare attitudes, over their life course, with some special families gradually accepting their specialness as a Shidu family and, at the same time, rebuilding their identity as normal older people. In the early stage of Shidu, special families tended to be more inclined to spend time together in similar groups, while in the middle and late stages of Shidu, their social support networks expanded and their identity diversified, so that they were not only special families, they also were part of many ordinary older age groups. This change revealed from their welfare experiences was not only a continuation of the special family’s life course, but also a reflection of their improved mental and physical health and the effectiveness of the welfare provisions they were receiving.


*In fact, at the beginning, these families are inferior, [and] it is certainly more appropriate for such people to be together. They are in the same situation, they can share the pain, there is also more common language, they also huddle for warmth. In the healing process, the real wound healed, they can gradually get over this hurdle.*

*(Interviewee DQZF-1)*


As discussed earlier, the specialization and de-specialization dimensions of welfare experiences can coexist within the special families. Special families wanted to both make a difference in and gain a positive influence from normal social networks, while they also regarded themselves as essentially different from other older adults and feared that their negative emotions might become a burden to others. Trying to join new groups is an important step in expanding their social support networks, but the social stigma of being a special Shidu family is invisibly internalized [44]. One particular family member (DDJF-3) said: *“It works to go out all day long, people are not in the same situation as me, [and] I don’t bring my kind of unhappiness to that group of people, who can influence me––I’m happy when people are happy, but when I’m not happy, I go home. If I’m not happy, I’m not going to influence people”.*

## 5. Discussion and Implications

This study adopted a qualitative research method to analyze the welfare experiences of special families in Jinan city, Shandong Province. The findings of the study were based on a generalized analysis of interviews with special families and related service providers, which connected the specialization dimension of welfare experiences with identity-oriented, targeted, and comprehensive characteristics, and the de-specialization dimension of welfare experiences with identity-denied, excluded, and hidden characteristics. The study also revealed the complexity of welfare experiences and the dynamics between those two dimensions in different special families, different family members, and different life-course periods. Some special families and some members of special families preferred more specialized welfare benefits which were provided only for them, some special families and some members of special families preferred more de-specialized welfare benefits which were provided for them not just because they were special families but because they were people who needed the benefits, and some special families and some members of special families wanted to receive specialized welfare benefits while refusing to be labeled or stigmatized. There were some special families or members of special families who gradually shifted from the specialization-characterized welfare attitudes to de-specialization characterized welfare attitudes over their life course. The following discussion and implications from the study’s findings are categorized into the theoretical and practical domains. 

In the theoretical domain, the positive social welfare theory from Giddens [45] suggests that “no responsibility means no rights”, and holds that responsibility and rights are inseparable. In our study, as beneficiaries of welfare benefits, special families were labeled as “lost and alone”, which lead to a distinctive focus of attention on them from the outside world. That excessive attention was a manifestation and unintended consequence of the alienation of these special families’ social rights, and it was different from the responsibilities that they were supposed to assume and their own initiative that they were expected to bring into play. Although the different tendencies that accompany specialization and de-specialization are subjective interpretations at the individual level, the stigmatization brought about by the label of “lost only child” is a construct of the mainstream social discourse and cultural environment. Furthermore, the stigmatizing consequences of another label, “dispossessed”, are constructs of the dominant social discourse and cultural environment, and have proven to be a transmissible social process [46]. The dynamics between the specialization and de-specialization dimensions and their effects on special families is a social problem arising from the operation of social mechanisms and welfare policies, rather than a personal problem of special families. This study attempted to understand the different characteristics of the two dimensions, and to elucidate the inherent tension between the assignment of a specialized status and that of a de-specialized status, which is better understood as a complex social problem of social welfare policies for disadvantaged groups, involving the concepts of rights, labeling, and stigmatization, than as simply negative consequences for special families themselves. 

Moreover, the stigmatization of this group should be eliminated on the macro social level. In the process of realizing their rights, special families are forced to bear the consequences of being stigmatized, and this additional result of alienation is neither within the scope of responsibility they should bear, nor should it exist. After the loss of their only child, some nuclear families face a crisis of disintegration, while relatives and friends gradually reduce their frequency of communication, social support networks weaken, and social stigmatization leads to both a real life crisis and a psychological crisis. 

In the practical domain, the policy makers and service providers should realize that while supporting special families, the implementation of social welfare policies may also label and stigmatize special families and become structural and cultural factors that cause the marginalization and isolation of special families from society. The government should optimize the welfare services designated for the special families, including offering them additional professional service providers, more elaborate service content and processes, and more refined support services. 

First, we should optimize social welfare policies for special families by taking into account their specialized and de-specialized welfare needs. The policy for special families should fully consider their specialized and de-specialized needs, and the tension between the two dimensions. It should not only provide policies and services to meet the specialized needs of special families, but also provide support and services to meet their de-specialized needs, so as to help them overcome the pain of the death or severe disability of their only child and return to normal life. Second, the process of providing social welfare services to special families should be improved to meet their differentiated specialized and de-specialized welfare needs. Service providers for special families should have a full understanding of, and be sensitive to, their specialized and de-specialized needs, and the tension between the two dimensions. In the process of providing services, they should provide targeted specialized or de-specialized services for different special families and family members, special families in different living conditions and different life course to meet their differentiated specialized or de-specialized needs. Third, a related policy package should be provided to the special families as instrumental guidance, including special social welfare policies available only to them and general social welfare policies for their age groups, and follow their autonomies to choose to receive specialized and de-specialized social welfare services when they need them.

This research had certain limitations. First, the study was conducted in Jinan only, and mainly in urban communities, so it cannot reflect the nationwide situations of community welfare experiences of special families, and their needs and interpretations, especially in rural communities. Second, the issue of special families under the one-child policy is a unique problem, to a degree, and the findings presented were based on an understanding of both the specialization and de-specialization dimensions of the families’ subjective experiences. Less consideration was given to theories about losing one’s child in normal situations, and that perspective should be included in further studies. 

## Figures and Tables

**Table 1 ijerph-20-04327-t001:** Basic information of the interviewees.

Special families	16 main research participants met the sampling criteria: living in Huaiyin district as citizens; who lost their only child.
Related welfare providers	7 social workers who worked with the special families; 4 related staff members of the District Civil Affairs Bureau; 4 managers of a planning family support committee for the streets and community; 2 interviewees in charge of special family support from the National Health Commission.

## Data Availability

Not applicable.

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
