# Peer review of "Exploring the Dynamics between Specialization and De-Specialization of Welfare Experiences: A Qualitative Study of the Special Families under the One-Child Policy in China"

_ijerph, 2023, doi:10.3390/ijerph20054327_

Round 1

Reviewer 1 Report

Special families in mainland China is an important topic and there is a lack of evidence and discussion on welfare experience of special families in existing research. Findings from the study provided useful evidence.

However, this paper itself lacks rigorous rationale of the definition, the usage of special family and Shidu family is mixed, literature review on special families or related families in other countries is quite limited and the discussion section needs some more deep thinking. 

Thank you for the opportunity to review this manuscript. Special families in mainland China is an important topic and there is a lack of evidence and discussion on welfare experience of special families in existing research. Findings from the study provided useful evidence.

However, this paper itself lacks rigorous rationale of the definition, the usage of special family and Shidu family is mixed, literature review on special families or related families in other countries is quite limited and the discussion section needs some more deep thinking. The detailed comments are as below:

1.      In Introduction section, para1, it requires more explanations of the usage of special families and why not to use the definition Shidu families and families with the only child has a severe disability. It seems that the concept of special families is defined officially rather than academically and experiences and trauma of Shidu family and families in which the only child has a severe disability is evident different, so it is important to explain why these two types of families should be integrated and analyzed, and the significance academically?).

2.      The usage of special family and Shidu family is mixed throughout the paper, which makes the statement and the explanation unclear, for instance the statement in 2.3 section: “Studies have pointed out that some special families are more resistant to their special Shidu identities…”. According to the definition of this study, families in which the only child has a severe disability is included in special families, but it is not belong to the Shidu family, which makes this statement not very rigorously. This problem also in finding section.

3.      In Literature Review 2.2 section, the existing evidence focused on the situation of welfare policies for special families, yet lack of the statement of the impact of policy and related factors (such as cultural, social factors which were mentioned in discussion section) on the experiences and interpretation of special families. Such statement is mentioned in discussion section but not mentioned enough in literature review. More evidence should be reviewed so that it can be reposed in discussion section rigorously to make this study more rigorously.

4.      In Literature Review 2.3 section, para 2, the statement of “their marginalization included both structural and psychological dimensions” is a good point, however, it seems that the evidence of this statement is occupied by psychological dimensions but little by structural dimensions. This problem is also existed in discussion section.

5.      In Literature Review section, although especial families under one child policy is unique in China, the evidence of lost-single-child family in other countries is also existed and should be reviewed. This problem is also influenced the deep thinking of discussion section.

6.      Methodology is questionable, in addition to the unclear definition of special families, it requires more explanation on the date collection from the participants of the providers of related welfare, as the research question is “to present the subjective narratives ..of welfare experiences of those special families”.  Why it should collect date from welfare providers, and what information should be collected which associated with the subjective experience of special families?

7.      In Findings section, as mentioned before, the experiences and trauma of Shidu and special families are different, it suggests to indicate the evident characteristics of these two types of families rather than the mixed statement, especially when described the identity of special families. Such problem is still due to the lack of rationale of the usage of special families from the academic perspective.

8.      In Discussion and Implication section needs some more deep thinking, for instance, more discussion on the structural and cultural influence should be supplemented.

9.      In Discussion and Implication section, para 4 page 10, it is better to indicate a definite approach (integrated or separate strategy to the existing welfare system which) so that it can make the suggestions more feasible, as it related to response the key point findings of this paper –“dynamics between specialization and de-specialization”(4.3 section), and this findings also implies a dilemma: how to deal with the difficulties of targeted services to the special families and meanwhile the reduction of the effects of stigmatization (universal services may be more suitable) according to the findings of this study?

Reviewer 2 Report

With the hope of major revision, I raise fundamental questions.

  1. China has let couples have up to three children recently. Under the new policy, why is your study still meaningful? In 4.3.1., you wrote aging of the population makes the one-child policy abolished. But the real problem is the low birth rate. Please contemplate how your study can embrace the dynamics. 
  2. What are your research questions? How could the literature review help construct them? It is unclear what your study brings new findings. It would help to organize a better literature review to lead to your research. 
  3. (Introduction) Consider restructuring your introduction. The current introduction needs clarification about what you want to accomplish. Why did you choose Jinan city, Shandong Province? Does this area have more special families? Where is it in China? Remember that many of your readers have a little background about China, and your introduction should guide the rest of your study. If your opening is poor, your readers won't read any further. 
  4. (Demographics) What is your sampling strategy? How are 33 study participants/samples comprising of? Please provide more information and summary statistics. Why and how did you choose each participant? Is there any IRB protocol for human subject study? How confident are you in generalizing your findings?
  5. (Research Methods) What are your qualitative research methods? Did you have coding strategies? Were your questions structured, semi-structured or open-ended? 
  6. (Theory) You introduced the positive social welfare theory at almost the end of the manuscript. The theoretical background should be earlier to guide your study. What is the positive social welfare theory? Your explanation needs to be sufficient to relate to your study findings. 
  7. (Structure) If your findings are in section 4, where are section 5: conclusions and section 6: implications? 
  8. (Template) MDPI journals have been using their own templates. I suggest you use it. You will realize that the template requires a different reference format, for example. 
  9. (Conclusions) Please consider my points 1-8 for better conclusions and implications. "Support services and better community activities" are too general to conclude your research.  

Round 2

Reviewer 2 Report

I get the difference between a new policy (having up to three children) and a one-child policy. But my first question was why special families matter despite this change. There may be literature about your study, and I request you explain from the literature review.

Your literature review and theory are for sound research questions and hypotheses.  

Please remember hundreds of thousands of readers who want to know why your study is important and worth reading.

Research questions you brought:

1.     What are the experience of having received the special family support policies?

2.     Do they want special arrangements or do they want to dilute their special status?

3.     What are the dynamics between specialization and de-specialization when obtaining welfare support services?

Consider your research questions based on the rule of thumb:

1.     What do you want to discover?

2.     What is the purpose of your study?

3.     What is the phenomenon or phenomena?

What is your research method? I get a purposive sampling from 33 special families in the Huaiyin District, Jinan City. You used open coding. Yes, I get that, as well. But they are part of your research method. There are reasons, aims, and processes for open coding. Please elaborate on your research method.     

Why and how are Table 1 helpful? I get you guarantee the anonymity of each sample. But why and how is Table 1 useful other than this? If it were quantitative analysis, this could be summary statistics. Summary statistics deliver demographic information. Please consider the use of Table 1. This Table is unnecessary.

I tried to match your findings to your research questions. It is imaginable that losing an only child is tragic and takes a long time to recover. If you argue that is your primary finding, it is reasonable but too obvious (duh). You define de-specialization as identity-denied, excluded, and hidden features from subjective welfare experiences. I don’t understand this. Are you saying those de-specialized cannot recover from losing child? Or are you saying those special families cannot get support from the government? And my primary question goes to how and what welfare services could go to special and de-special families?

Round 3

Reviewer 2 Report

As a peer reviewer, I commented as much as possible twice for a better manuscript. I hope my comments help you improve your manuscript. 

I appreciate your reflections and no more requests. But once again, please proofread carefully for publication.